# DLBCL 1L—What to Expect beyond R-CHOP?

**DOI:** 10.3390/cancers14061453

**Published:** 2022-03-11

**Authors:** Maike Stegemann, Sophy Denker, Clemens A. Schmitt

**Affiliations:** 1Department of Hematology and Medical Oncology, Kepler University Hospital, Johannes Kepler University, Krankenhausstraße 9, 4020 Linz, Austria; maike.stegemann@kepleruniklinikum.at; 2Charité—Universitätsmedizin Berlin, Corporate Member of Freie Universität Berlin and Humboldt-Universität zu Berlin, Medical Department, Division of Hematology, Oncology and Tumor Immunology, Campus Virchow-Klinikum, Augustenburger Platz 1, 13353 Berlin, Germany; sophy.denker@charite.de; 3Berlin Institute of Health at Charité—Universitätsmedizin Berlin, Charitéplatz 1, 10117 Berlin, Germany; 4Max-Delbrück-Center for Molecular Medicine, Helmholtz Association, Robert-Rössle-Straße 10, 13125 Berlin, Germany

**Keywords:** 1L therapy, bispecific antibodies, CAR-T-cells, diffuse large B-cell lymphoma, immuno-oncology agents, targeted therapeutics

## Abstract

**Simple Summary:**

Diffuse large B-cell lymphoma (DLBCL) is the most common aggressive non-Hodgkin’s lymphoma. About two-thirds of patients are cured by the first-line (1L) standard of care (SOC), the R-CHOP (Rituximab, Cyclophosphamide, Doxorubicin, Vincristine and Prednisolone) immunochemotherapy protocol. The profound molecular heterogeneity of DLBCL is the underlying reason why many patients, despite improved next-line options, eventually succumb to the disease. Hence, enhancing the efficacy of 1L treatment is critical for improving long-term outcomes in DLBCL. A plethora of novel treatment options with potential in later lines is currently under evaluation in 1L settings. We summarize here the established and emerging strategies for newly diagnosed DLBCL and emphasize the need for individualized treatment decisions.

**Abstract:**

The R-CHOP immunochemotherapy protocol has been the first-line (1L) standard of care (SOC) for diffuse large B-cell lymphoma (DLBCL) patients for decades and is curative in approximately two-thirds of patients. Numerous randomized phase III trials, most of them in an “R-CHOP ± X” design, failed to further improve outcomes. This was mainly due to increased toxicity, the large proportion of patients not in need of more than R-CHOP, and the extensive molecular heterogeneity of the disease, raising the bar for “one-size-fits-all” concepts. Recently, an R-CHP regimen extended by the anti-CD79b antibody–drug conjugate (ADC) Polatuzumab Vedotin proved superior to R-CHOP in terms of progression-free survival (PFS) in the POLARIX phase III trial. Moreover, a number of targeted agents, especially the Bruton’s tyrosine kinase (BTK) inhibitor Ibrutinib, seem to have activity in certain patient subsets in 1L and are currently being tested in front-line regimens. Chimeric antigen receptor (CAR) T-cells, achieving remarkable results in ≥3L scenarios, are being exploited in earlier lines of therapy, while T-cell-engaging bispecific antibodies emerge as conceptual competitors of CAR T-cells. Hence, we present here the findings and lessons learnt from phase III 1L trials and piloting phase II studies in relapsed/refractory (R/R) and 1L settings, and survey chemotherapy-free regimens with respect to their efficacy and future potential in 1L. Novel agents and their mode of action will be discussed in light of the molecular landscape of DLBCL and personalized 1L perspectives for the challenging patient population not cured by the SOC.

## 1. Introduction

The standard of care (SOC) in first-line (1L) therapy of diffuse large B-cell lymphoma (DLBCL) remains the R-CHOP regimen (Rituximab, Cyclophosphamide, Doxorubicin, Vincristine and Prednisolone), achieving cure rates of about 60–70% across all patients with 6–8 (and in selected settings even fewer) cycles of therapy [1,2,3].

DLBCL is the most frequent aggressive non-Hodgkin’s lymphoma (aNHL), with an incidence of seven cases per 100,000 people per year. The majority of patients are diagnosed in their seventies or above, but disease onset can occur at any age [4]. Traditionally, individual risk is stratified by the international prognostic index (IPI), including age, overall performance status (i.e., Eastern Cooperative Oncology Group [ECOG] performance status), Ann Arbor stage, involvement of extranodal sites, and the serum level of lactate dehydrogenase (LDH) to estimate progression-free and overall survival (PFS, OS) rates [5]. As all age groups can be affected by DLBCL, and increased age is associated with inferior outcome, IPI has been adapted to the age-adjusted IPI (aaIPI) as an age-independent survival score. Four risk groups (low, low–intermediate, high–intermediate, and high) with decreasing five-year OS rates (83%, 69%, 46%, and 32%, respectively) were defined [5]. Disappointingly, no profound improvement in long-term outcome, namely OS, has been achieved in the R-CHOP era so far. Numerous molecular subgroups with enhanced risk of failure, such as the presence of MYC, BCL2, and/or BCL6 overexpression or translocation, the activated B-cell (ABC) subtype as cell of origin (COO), or an immunoblastic or plasmablastic morphology, have been known for quite some time. More recently, more refined subtyping was achieved by in-depth molecular profiling of large DLBCL case numbers to better predict biologically any inferior outcomes of 1L R-CHOP therapy [6]. In addition to these risk factors, the overall condition and comorbidities of the patients are important determinants that impact on treatability, anticipated treatment outcome, and the specific treatment options to be chosen.

An inherent barrier to superior long-term outcomes in DLBCL is the profound molecular heterogeneity, not mirrored by the histological uniformity of the diffuse blastoid morphology across patients. With the advent of next-generation sequencing (NGS), allowing extensive (epi-)genome and transcriptome analyses [7,8], DLBCL can be subdivided into a plethora of novel subtypes. Foremost are the molecular subgroups and clusters, as inaugurated in recent years by the Staudt and Shipp laboratories [9,10]. Pointing to molecularly informed, subtype-specific strategies in the future [11,12], these findings dramatically underscore how unlikely the addition of a single targeting agent is to improve outcomes of these highly diverse lymphoma biologies in “all-comer” trials.

## 2. Deescalating or Reducing the SOC

Six, rarely eight, cycles of R-CHOP therapy (typically every three or, in some settings, every two weeks) are administered to newly diagnosed (nd) DLBCL patients as SOC [13,14,15,16]. The efficacy of the SOC in terms of complete response (CR) rates, PFS, and OS mainly depends on risk factors such as the aaIPI score, disease stage, and molecular make-up (e.g., “double-hit” lymphoma, as discussed below) [13,14,16]. With PET-guided response assessment in routine practice [17], it is now broadly accepted not to apply consolidative radiation therapy to bulk manifestations if the lesion is PET-negative at the end of treatment (EOT) [18,19]. Deescalating the SOC to four cycles in younger patients (18–60 years) with no risk factors, i.e., aaIPI 0, stage I/II without bulky disease, normal LDH and ECOG 0–1, as learnt from the German FLYER study, is not inferior to six cycles [13]. Moreover, R-CHOP in chemoreduced dose intensity (“R-miniCHOP”) is widely used for less fit, elderly patients, especially above the age of 80, to limit toxicity, while accepting somewhat lower efficacy, although no formal head-to-head comparison has been conducted with the SOC [20].

## 3. Intensified Chemotherapy Regimens Lead to Good Efficacy at the Cost of High Toxicity

Increased treatment intensity with eight instead of six cycles of R-CHOP in high-risk disease produced an inferior outcome [16,21]. The attempt to replace Rituximab with Obinutuzumab, a second-generation, Fc-glycoengineered CD20 monoclonal antibody, also failed. No improvement in OS was seen in the large, randomized phase III GOYA trial when comparing eight cycles of Rituximab or Obinutuzumab (G) with eight or six cycles of R- or G-CHOP [16]. The extension of the chemo backbone by Etoposide (R-CHOEP) is often applied in high-risk disease and was established in the pre-Rituximab era. Large retrospective analyses suggested the superiority of R-CHOEP (with slight variations also known as R-EPOCH) in young high-risk patients, especially in the subgroup of germinal center B-cell (GCB) origin, with a moderate increase in toxicity [22,23,24]. Earlier data pinpointed a less favorable efficacy and toxicity profile of CHOEP compared to CHOP in elderly patients on a biweekly schedule [25]. More recently, a large, randomized phase III trial including 524 patients with stage ≥ II disease failed to show the superiority of dose-adjusted (DA-)R-EPOCH compared to R-CHOP. Even though the mortality rate was similar in both groups, DA-R-EPOCH was associated with higher toxicity, leading to a completion of the protocol by only 83% [26]. The remarkable outcome of younger patients at high risk (18–60 years, aaIPI 2 or 3) in the R-CHOEP control arm of the German R-MegaCHOEP trial implied the potential superiority of R-CHOEP in such a population, as backed by Swedish registry data, but a direct comparison to the SOC is missing [27,28]. However, these higher-risk and, thus, often higher tumor volume-bearing patients received R-CHOEP for eight rather than six cycles, so were possibly in need of more or longer induction therapy as compared to low-burden patients. In a retrospective analysis of a high-risk DLBCL collective bearing MYC rearrangements as single, double or triple events (the latter in conjunction with BCL2 and/or BCL6 translocations), no benefit of R-CHOEP over R-CHOP was detected [29]. The intensified chemotherapy regimen R-ACVBP (Rituximab, Etoposide, Doxorubicin, Cyclophosphamide, Vindesine, Bleomycin, Prednisolone, Methotrexate intrathecally and intravenously, Ifosfamide, and Cytarabine) is, so far, the only protocol that has outperformed R-CHOP in terms of PFS and OS when tested in a randomized fashion in younger patients (<60 years of age) with intermediate-risk disease (aaIPI = 1) [30,31]. However, those results were not confirmed by the recent “GAINED” study, which showed a similar outcome when comparing CHOP with the ACVBP backbone plus a CD20 antibody in a more complex setting [32]. Moreover, due to its enhanced toxicity, ACVBP is not widely used [31]. Numerous attempts have aimed to improve outcomes in high-risk patient subsets through upfront high-dose chemotherapy with autologous stem cell support, among them the failed R-MegaCHOEP trial [27]. An Italian phase III trial testing R-CHOP-14 vs. two types of intensified high-dose chemotherapy plus autologous stem cell transplantation-based protocols found superior failure-free survival for the high-dose arm. However, no benefit in OS was observed, thus not supporting a more intense strategy upfront [33].

Intensification of Rituximab on top of the CHOP chemotherapy backbone in elderly patients (DENSE-R-CHOP, SMARTE-R-CHOP) or on the CHOEP backbone in high-risk younger patients in the DENSE-R-MegaCHOEP trial did not provide superior outcomes in terms of OS [34,35,36,37]. Dose intensification of Rituximab was associated with significantly more treatment-related complications, especially neutropenia and infections, in comparison to SOC [34]. Hence, these findings regarding chemo backbone and/or antibody intensification are generally not in favor of an a priori treatment-escalating strategy for DLBCL patients at particular risk at diagnosis.

However, data are more controversial and difficult to interpret regarding interim PET (iPET)-based (conducted after two to four cycles of an R-CHOP-like induction) intensification of the chemotherapy regimen for patients with no or a slow metabolic response at this time. The German PETAL trial failed to demonstrate a benefit to immediately switching iPET+ patients, after two cycles of R-CHOP, to the more intense Burkitt lymphoma protocol [38,39]. In contrast, generally remarkable outcome data were reported from the French GAINED study. Patients with iPET positivity after two cycles (iPET2) who reached a complete response (CR) after four cycles of therapy received treatment intensification with upfront autologous stem cell transplantation, resulting in similar OS rates between iPET 2-positive and -negative patients. PET-suggested inferiority, partially equalizing long-term outcome data, which are instrumental in identifying patients with a positive interim PET after four cycles of induction therapy as a true high-risk collective [32]. In essence, early-intensified chemotherapy induction may exert enhanced anti lymphoma activity, but at the price of increased toxicity, reducing potential benefits at least in part due to the risk of subsequent toxicity-related dose delays [25,26,32] (see Table 1). Given the profound molecular heterogeneity of DLBCL, as alluded to above, it is fairly unexpected that we will overcome insufficient R-CHOP sensitivity by a mere dose-escalation strategy. It rather seems that molecular vulnerabilities or immune therapeutic targets deserve in-depth exploitation in susceptible subgroups.

## 4. Clinical Outcome Predicted by the COO Classification, or a MYC/BCL2-Focused Expression or Translocation Status

DLBCL is a heterogenous disease that can be further classified with numerous criteria. The COO classification is widely used to subdivide DLBCL into GCB and ABC subtypes, using a transcriptome-based algorithm, and approximated by routinely available immunohistochemistry (IHC)-based marker panels. Next to the widely used Hans classifier [50], the Choi classifier [51], as well as three- and four-marker algorithms (Visco–Young algorithm) [52], have been established to discriminate GCB from non-GCB cases by IHC. All these tools are based on CD10 and BCL6; the Hans and Choi classifiers further use MUM1/IRF4 [50], GCET1 and FOXP1 [51], while the Visco–Young classifiers include FOXP1 (a three-marker assay) and GCET1 (a four-marker assay) [52]. Although those COO designators show reasonable concordance with gene expression profiling (GEP), the accurate recapitulation of the prime biology/outcome linkage—the superiority of the GCB subtype to R-CHOP SOC—has been questioned with IHC-based tools. Also, those merely diagnostic tools are not suited to provide deeper insights into the underlying signaling networks. Originally, the COO classification reflected a transcriptome-based clustering method that assigns the GCB and ABC states to a certain probability via a linear predictor score. This calculation summarizes the expression levels of a distinct set of genes. This explains, at least in part, why IHC-based algorithms can only serve as an approximation of bona fide transcript-based COO classifiers and thus function as COO-related risk stratifiers or outcome predictors [53]. GEP-based COO classifiers—for example, the Lymph2Cx assay using NanoString oligonucleotide hybridization technology [54,55], RNA microarrays [56,57], the Illumina DASL (cDNA-mediated Annealing, Selection, extension and Ligation) assay [43], or HTG EdgeSeq technology [44]—lead to reliable COO designation into GCB and ABC subtypes, with high concordance across the different platforms [7,58,59]. While IHC is easy to apply in routine pathology, and, hence, widely used in daily practice, any IHC-based clinical decision should be taken with caution. The disparity between IHC- and transcriptome-based methods can be significant, potentially affecting study results. Accordingly, most COO-sensitive trial protocols, at least for large phase III studies, rely on transcript-based COO assays.

Different subclassifications that require additional fluorescence in situ hybridization (FISH) have been introduced, aiming to specify patients with an increased risk profile. Besides double-hit (DH) or triple-hit (TH) lymphomas, referring to translocations involving the MYC and BCL2 and/or BCL6 genes detected via FISH, the double-expressor (DE, assessed by IHC) lymphoma phenotype refers to strongly detectable protein expression of the MYC and BCL2 gene products. DH-, TH-, and, as they were previously named, “Burkitt-like-” lymphomas were summarized as high-grade B-cell lymphoma (HGBL) by the WHO in 2016 [6]. Entities without MYC and BCL2 and/or BCL6 rearrangement but that appear blastoid or in between Burkitt’s lymphoma and DLBCL are defined as HGBL if not otherwise specified (HGBL-NOS).

Although patients with DH lymphomas showed inferior PFS and OS when treated with R-CHOP [60], the prognostic impact of MYC and BCL2 (over)expression in DLBCL in DE lymphomas and HGBL-NOS is still not clear [61,62,63,64,65]. Unfortunately, all trials designed to effectively overcome the negative prognostic impact of HGBL via intensified chemotherapy failed. Promising but yet immature data on immuno-oncology agents in high-risk situations will be further discussed in this review. 

## 5. Polatuzumab Vedotin Extension of the Immunochemotherapy Backbone as a New 1L SOC?

Polatuzumab vedotin is an antibody–drug conjugate (ADC) targeting CD79b, a B-cell receptor (BCR) complex-associated molecule. After internalization of the ADC followed by intracellular cleavage, its payload monomethyl auristatin E, an antimitotic agent, is released and triggers apoptosis. Approved in combination with Bendamustin and Rituximab for R/R DLBCL [66], a randomized phase III trial testing Polatuzumab vedotin in combination with Rituximab and CHP (Cyclophosphamide, Doxorubicin and Prednisone) in nd DLBCL (IPI ≥ 2) vs. R-CHOP was conducted and recently reported [47]. The study, named POLARIX, met its primary endpoint, i.e., the Polatuzumab arm was superior regarding PFS. Specifically, Polatuzumab produced a superior PFS with a hazard ratio of 0.73 (95% CI 0.57–0.95) and a 2-y PFS advantage of 6.5% [47]. The two arms were well balanced, and the overall collective certainly not diluted by too many low-risk patients. No differences in overall response, i.e., CR and PR rates, and regarding OS were observed. It remains unclear why PFS improvement failed to translate into superior OS, as typically seen in 1L therapies for DLBCL [67]. One possible reason is the harder-to-salvage biology in failures following Polatuzumab-R-CHP, perhaps due to a better debulking capacity of cellular components, while sparing the putative cancer stem cells responsible for disease relapse. It also seems that, due to the PFS definition, unplanned consolidative radiation may have contributed to equalized OS—decreasing the presumed benefit regarding quality of life due to prevented relapses. More complex effects, including targets in the lymphoma microenvironment and the immune system, may underlie the reported results. Key hints came from the subgroup analyses, which indicated that older (>60 years) males with higher-risk disease (i.e., IPI 3–5) seem to particularly benefit from Polatuzumab, as, most notably, patients with an ABC subtype and a DE, but not DH, molecular profile did [47]. Even with a longer follow-up, the overlap of the OS curves makes it unlikely that a meaningful survival benefit will emerge later. Additional toxicities by the ADC were moderate, and dose adherence was slightly higher in the Polatuzumab group.

Whether Polatuzumab-R-CHP should be considered the new SOC is currently a major debate in the field. Although producing a PFS benefit with little extra toxicity, the bar to replace the long-term standard appears too high for a regimen not enhancing the CR rate and not prolonging OS. More detailed clinical and molecular analyses are needed to draw more definitive conclusions from the subgroup findings mentioned before, and to see whether certain subgroups, on the contrary, may even experience a potential disadvantage when treated with Polatuzumab-R-CHP instead of R-CHOP. Beyond the additional costs, a move of this agent to 1L might also deprive us of a powerful compound in later lines of therapy, especially for high-dose treatment/autotransplant-ineligible patients and those in need of an effective bridging strategy to chimeric antigen receptor (CAR) T-cell therapy. Given its mode of action as a binder to a ubiquitously expressed surface receptor, one would expect Polatuzumab to be less subgroup-selective than targeted therapeutics, and hence to exert its activity across all DLBCL patients—among them, many not in need of more than R-CHOP. The first subgroup analyses, however, do not seem to support this view. Thus, we look forward to biomarker studies that will hopefully pinpoint a sizeable subgroup of nd DLBCL patients with an unmet need who experience not only a PFS but OS benefit due to the Polatuzumab extension. It seems that such a biomarker has not been clearly defined yet, but POLARIX and other, formally negative phase III 1L trials in DLBCL teach us how important integral biomarker studies are. Future studies may show whether different risk scenarios, such as slow metabolic responses identified at interim PET imaging or delayed clearance of lymphoma-specific circulating DNA, are good indications for a Polatuzumab-based augmentation of the current induction therapy.

## 6. Targeted Therapies Addressing the BCR/NF-κB Pathway

Constitutively active BCR/NF-κB signaling reflects a key survival pathway in B-cell malignancies in general and DLBCL in particular [68]. The signaling activity is typically more pronounced in the ABC subtype. In the ABC subtype, the pathway is chronically active due to auto-antigenic stimulation of the BCR, auto-activating BCR clustering or enrichment for BCR/NF-κB activating mutations, e.g., at the level of the BCR complex-associated CD79a/b proteins, CARD11 or the tumor necrosis factor-α (TNFα)-induced protein 3 (TNFAIP3)/A20, collectively driving high-level NF-κB activity [69,70,71]. In addition, the NF-κB pathway is activated via Toll-like receptor signaling through the adaptor molecule MyD88 [9,10,72]. Activating mutations in the Toll/Interleukin 1 receptor domain of MyD88 are predominantly found in ABC lymphomas [69]. BCR/NF-κB signaling is of importance in the GCB subtype as well; however, its tonic activity is less intense, less frequently due to activating mutations, and executed more selectively via the PI3 kinase/Akt/mTOR downstream signaling cascade [71]. Hence, numerous studies have focused on the NF-κB pathway in DLBCL patients of the ABC subtype. Three candidates have been tested in combination with R-CHOP: the BTK inhibitor Ibrutinib, the proteasome blocker Bortezomib, preventing the degradation of the NF-κB antagonist IκB, and Lenalidomide as an immunomodulatory drug assumed to inhibit downstream NF-κB signaling. 

In a phase I/II study for relapsed or refractory (R/R) DLBCL patients, Bortezomib in combination with DA-R-EPOCH produced significantly better outcomes in the ABC subgroup compared to the GCB subtype. Unlike the inferiority to 1L therapy in nd ABC DLBCL, no COO-related differences were observed in response to standard salvage therapy [73]. However, the promising results of a phase I/II study investigating the effect of Bortezomib + R-CHOP in previously untreated non-GCB DLBCL patients [74] could not be confirmed in a randomized phase II trial [75]. Moreover, REMoDL-B, a large, randomized phase III trial investigating the efficacy of R-CHOP ± Bortezomib, failed to show superiority after the addition of Bortezomib [43] (see Table 1).

When exploited in R/R DLBCL, single-agent Ibrutinib produced particularly encouraging signals in an ABC subgroup with MyD88-mutant disease [76]. The large, randomized phase III 1L study PHOENIX compared the R-CHOP backbone with or without Ibrutinib, intended for non-GCB subtype patients only (for study entry COO assessed by IHC, and to be reassessed as ABC or GCB by a more refined transcript-based method later). However, among the 838 patients included, no survival differences could be observed among the treatment cohorts. Nevertheless, preplanned subgroup analyses revealed significantly better outcomes in patients younger than 60. Molecular reevaluation of the COO by GEP showed that not only ABC- but quite a number of GCB-subtype patients were enrolled, with the Ibrutinib benefit particularly visible in young ABC patients. The benefit to younger patients was outweighed by the inferior outcome of the elderly participants, who experienced, in the absence of mandatory prophylaxes, increased toxicities, especially in the Ibrutinib arm. Therefore, the R-CHOP backbone could be administered only at profoundly reduced dose adherence [44]. A retrospective in-depth analysis of the genetic subtypes in the PHOENIX trial, according to the new molecular subgroups recently determined by the Staudt laboratory [9], revealed the importance of three previously defined genetic subtypes regardless of the COO [12]. An impressive improvement in three-year event-free survival (EFS) from ≤50% to 100% by the extension of Ibrutinib to R-CHOP was seen in younger patients (<60 years) belonging to the MyD88 mutation-governed MCD or the Notch1-activated N1 subtypes [12]. Further investigations are needed to confirm these promising results in a prospective setting. Moreover, Ibrutinib-R-CHOP also proved highly effective in DLBCL, with high-risk “double expressor status” in another retrospective analysis of the PHOENIX trial. Adding Ibrutinib to R-CHOP neutralized the inferior EFS and OS outcome in younger patients with high co-expression of MYC and BCL2 (slightly differently defined from the more commonly used double-expressor lymphoma status; see above) [77]. Potentially addressing these points, “ESCALADE,” a phase III trial testing the addition of Acalabrutinib, a second-generation BTK inhibitor with enhanced target selectivity, in younger patients (<65 years) of non-GCB subtype, is currently recruiting [48] (see Table 1). 

The limited efficacy targeted approaches often achieve might be because collateral pathways are turned on in a compensatory fashion or downstream mediators gain activity. Therefore, double-targeting of the BCR/NF-κB cascade at a proximal and a distal point might lead to more lasting suppression of this critical signaling backbone—and is not necessarily restricted to ABC patients. Moreover, preclinical trials suggested a synergistic effect of BTK inhibition and proteasome blockade, even in previously Bortezomib-insensitive cell lines [78]. Based on these considerations, ImbruVeRCHOP [49] is another actively recruiting early-phase clinical trial that investigates the combined BCR/NF-κB blockade by Ibrutinib and Bortezomib added to the R-CHOP backbone in elderly (age 61–80) patients with higher-risk disease (defined as IPI ≥ 2) but no pre-selection according to COO (see Table 1). Tolerated quite well in the context of a quadruple growth factor, antibiotics and antiviral prophylaxis have shown encouraging interim results [49,79]. A unique feature of this trial is its complex molecular program, seeking transcriptional profiles not only prior to first drug exposure, but directly under cycle 1 and, depending on the remaining lymphoma burden, further along in the course of therapy, with the declared aim of extracting a gene signature that indicates benefit from R-CHOP plus Ibrutinib and Bortezomib, potentially independent of a COO designation.

Lenalidomide achieved significantly better OS in the ABC subgroup in R/R DLBCL when applied as monotherapy compared to a chemotherapy of the investigator’s choice [80]. Lenalidomide combined with Rituximab (called “R^2^”) has activity in R/R DLBCL [81]. Lenalidomide plus R-CHOP (“R^2^-CHOP”) was subsequently offered to nd DLBCL patients in two phase II trials, in which the addition of Lenalidomide neutralized the established ABC inferiority [82,83,84], but the consecutive randomized phase III “ROBUST” trial for ABC patients only [85] and R^2^-Mini-CHOP for nd DLBCL patients over 80 years [86] failed by producing higher toxicity without improving overall survival [85,86]. Furthermore, Lenalidomide demonstrated some efficacy as a maintenance treatment by prolonging PFS after successful induction therapy with R-CHOP, albeit without an OS impact, in a group of elderly patients (60–80 years of age) [87].

Notably, increased toxicity was a major obstacle in previous attempts to extend R-CHOP by additional agents. Beyond the studies discussed, Gemcitabine + R-CHOP, as an example, resulted in increased pulmonary toxicity [40], leading to the premature termination of the trial. The extension by Bevacizumab, a vascular endothelial growth factor (VEGF) inhibitor, provoked severe cardiac toxicity, such as congestive heart failure. Significant differences in outcome, even in a subgroup with confirmed VEGF activation, were not achieved [41,88]. Such additional adverse effects, or those that may be more frequent in a particularly vulnerable patient population, must be carefully considered and anticipated from earlier-phase signals, preventing phase III trials from failing for toxicity reasons or seeing a drop in dose adherence regarding the R-CHOP backbone. Although R-CHOP has been well established for decades, any additional drug must be taken with caution to avoid cumulative excess toxicity or unforeseen critical drug interactions. However, preemptive adjustments of dosing or even entire agents might also affect the comparability of the control and test arm in randomized settings. For instance, Vincristine, although frequently dose-reduced or omitted in the course of R-CHOP-based induction, is a core component of this well-established SOC. Eliminating Vincristine upfront in R-CHOP + X candidate regimens, such as the R-CHP plus Polatuzumab vedotin arm in the POLARIX trial, is understandable in light of the potential cumulative neurotoxicity conferred by both Vincristine and the ADC’s payload, but efficacy results must be interpreted accordingly [47].

In essence, the targeted agents or biologicals Bortezomib, Ibrutinib, and Lenalidomide failed in large, randomized phase III trials open to DLBCL patients based on COO entry criteria of varying robustness. Although results across all patients were disappointing, subsequent profiling of responsive versus insensitive lymphoma samples revealed remarkable clinical activity to these agents in combination with the SOC in distinct molecular subsets. Hence, these findings underscore the need for in-depth molecular scrutiny of all participants in large clinical trials to unmask, independent of met or missed primary endpoints, potential connections between genetic signatures and activities of candidate compounds. 

## 7. Extension of R-CHOP-like SOC by Other Small Molecules—New Hope and Failed Promises

In the GCB subtype, enhancer of Zeste homolog 2 (EZH2) promotes DLBCL development and exerts synergistic effects with BCL2 [89,90]. Encouraging preclinical results with Tazemetostat, a small-molecule EZH2 inhibitor, led to clinical trials [91]. Positive results were obtained with Tazemetostat as a monotherapy in R/R DLBCL [92,93], and it appeared to be safe in combination with R-CHOP in elderly nd DLBCL patients (age 60–80) with or without EZH2 mutation [94]. Currently, phase II trials are ongoing in EZH2-mutant DLBCL, as Tazemetostat was recently approved by the FDA for R/R EZH2-mutant follicular lymphoma [95].

Whether sole overexpression, unlike the MYC/BCL2 DE status, of the antiapoptotic regulator protein BCL2 indicates inferior long-term outcome in the Rituximab era is controversial [96,97,98,99]. However, BCL2 (over-)expression might be exploited by Venetoclax, a potent and highly selective, orally available BCL2 inhibitor. In the single-arm phase II CAVALLI 1L trial, Venetoclax was tested as an R-CHOP extension. Patients with nd CD20^+^ DLBCL, ECOG 0–2, IPI 2–5 and adequate organ function were included. Lymphoma material was analyzed regarding COO subtype, MYC*,* BCL2 and BCL6 expression (by IHC), as well as translocation status (by FISH). Matching previous reports, the proportion of BCL2^+^ cases (defined as ≥50% of the cells staining positive) was higher in the ABC subtype [7,60,64]. The addition of Venetoclax showed a tendency towards a superior PFS when compared to a matched R-CHOP control group of the GOYA trial, in BCL2^+^ subcohorts. Increased toxicity was a problem, accounting for dose delays [46]. With such vague efficacy but concerning toxicity results, the potential role of Venetoclax as an R-CHOP 1L extension is certainly not yet established. Notably, first results from a randomized phase II/III study, the ALLIANCE051701 trial, of DA-EPOCH-R ± Venetoclax in the high-risk arena of MYC/BCL2 DH lymphomas was presented at the ASH 2021 annual meeting [100]. The trial had to be prematurely closed because of overt toxicity in the Venetoclax arm, specifically increased hematological toxicity, a higher sepsis rate and more grade 5 adverse events (6/35 vs. 1/30), and, additionally, inferior outcome in terms of PFS and OS.

Exportin 1 (XPO-1) has a putative oncogenic function by exporting tumor suppressor proteins in different tumor entities and is associated with DH- and TH-DLBCL as well as impaired OS rates in DLBCL [101,102]. Selinexor, an inhibitor of the nuclear export protein Exportin 1 (XPO-1), has been investigated in an open-label phase 2b study, the “SADAL” trial, as a single agent for R/R DLBCL after two lines of treatment, where it showed an ORR of 28% (with 11% CR) [103]. Patients achieving a CR experienced durable responses. However, the study was designed to only include patients with slowly progressing, hence, “better-risk” relapse disease, which should be taken into consideration when interpreting these results. The preclinically reported efficacy of MYC^+^ XPO-1 expressing DLBCL [102] could not be confirmed in a post hoc analysis of the SADAL trial, and the outcome was independent of the COO [104]. DLBCL patients with MYC and BCL2 overexpression had a shorter OS (median of 5.1 vs. 13.7 months) and a lower ORR (14.8% vs. 46.2%) compared to normal expression levels [104].

## 8. Novel Antibody Targets at the Lymphoma Cell Surface

Tafasitamab, a CD19 monoclonal antibody with optimized target affinity and an engineered Fc portion to enhance antibody-dependent cellular cytotoxicity and phagocytosis (ADCC and ADCP, respectively), led to CR rates of around 40% in R/R DLBCL. The mechanism of action is based on the activation of both natural killer (NK) cells and macrophages, thereby possibly synergizing with Lenalidomide as an immune-modulating and presumably NK function-enhancing agent [105,106]. Based on these single-arm phase II “L-MIND” data [105,106] and a piloting phase Ib 1L study of this combination plus R-CHOP, termed “First-MIND” [45], the randomized “Front-MIND” phase III trial is now investigating the efficacy of Lenalidomide-R-CHOP ± Tafasitamab [107]. Safety will be of special interest since the L-MIND trial showed an acceptable, albeit not insignificant, toxicity profile [105,106]. In an interim evaluation of First-MIND, no concerning safety signals beyond the known toxicity profiles of R-CHOP or R^2^-CHOP were observed [45].

Loncastuximab tesirine, a CD19 ADC, together with Ibrutinib, presented an encouraging ORR of 73.7% (with 45.5% in CR) as salvage therapy for R/R DLBCL in the interim analysis of the actively recruiting phase II LOTIS-3 trial [108]. Responses appear to be more durable in the GCB subtype—making it even more interesting to speculate whether the CD19 ADC or the BTK inhibitor are more important here. The DLBCL patient cohort with a median age of 72 exhibited no overt safety signals so far. The data are yet immature but seem promising and this combination of CD19 ADC plus BTK inhibition bears potential for future 1L chemotherapy-free regimens. 

Another promising approach might be the addition of a PD1 or PD-L1 immune checkpoint blocker, such as Pembrolizumab, Nivolumab or Avelumab. While earlier single-agent investigations of immune checkpoint inhibition (ICI) across lymphoma entities in the R/R setting were rather disappointing [109,110,111], 1L combinations of the PD1 blocker Pembrolizumab or the PD-L1 blocker Avelumab with R-CHOP produced encouraging PFS signals [112,113]. The PD-L1 immune checkpoint blocker Avelumab is particularly interesting, since it may not only act via T-cell derepression but also subjects PD-L1-positive lymphoma cells to ADCC. In a pilot study, Avelumab and Rituximab were administered for two cycles prior to the start of R-CHOP. In this immunotherapy-only induction phase, a remarkably high ORR of 60% was detected, with PET-negative CR rates in 21% [113]. Although the data require verification, such a strategy might also be particularly interesting for patients who do not qualify for chemotherapy due to comorbidities.

CD47 blockade using Hu5F9-G4, a humanized IgG4 antibody binding CD47, in combination with Rituximab, is believed to restore the phagocytosis of tumor cells initiated by Rituximab. ORR and CR in patients with R/R DLBCL was 40% and 33%. Those results were obtained even though 95% of all patients were refractory to Rituximab before, underlining the synergistic effect of this combination [114]. To combine different immunotherapies and targeted therapies offers great potential; hence, studies testing rather underexplored agents in innovative combinations may open up novel directions for DLBCL treatment.

## 9. Chemotherapy-Free Induction Therapy Exhibits Promising Efficacy with Limited Toxicity

Chemotherapy-free induction therapy for DLBCL has been considered inefficient for too long. Combination therapies expanding on the R/R DLBCL-active Rituximab/Lenalidomide regimen [81] by the addition of Ibrutinib (“IR^2^”) produced surprisingly strong signals (ORR of 55–65%) in a phase I dose escalation and a subsequent phase II trial, reflected by a potential PFS/OS plateau of around 40% at 12 months. Both studies included patients with R/R DLBCL of non-GCB subtype and ineligible for high-dose chemotherapy [115,116]. In another phase II trial, termed “SMART Start,” patients with nd non-GCB DLBCL received two cycles of chemotherapy-free 1L induction therapy with the same agents, here abbreviated as RIL, followed by conventional immunochemotherapy with R-CHOP. An impressive ORR of 86% with CR rates of 36% were observed after two cycles of RIL [117]. The SMART Start concept with RIL, followed by a treatment escalation to RIL + EPOCH, is currently being evaluated in high-risk non-GCB DLBCL patients [118]. Based on those results, new trials should be implemented to explore the possibility of completely chemo-free induction therapy, specifically for patients who are unfit for conventional chemotherapy and enter an early metabolic CR after the first two cycles (see Table 2). Given the growing proportion of elderly patients with medical conditions and the increasing availability of non-chemo agents tested in novel combinations and sequences, a large amount of data is expected to be presented soon. Beyond the agents discussed in this and previous sections (e.g., ICI based on the PD-L1 blocker Avelumab, or the anti-CD79b ADC Polatuzumab vedotin), there are CAR T-cells and T-cell-recruiting bispecific antibodies pushing towards 1L. The latter (see below) already reported remarkable 1L activity as single-agent or in chemo-free combinations for medically less fit patients—but are highly promising for all DLBCL patients.

## 10. Maintenance Therapy Failed to Improve OS

An additional strategy to improve long-term outcomes after 1L therapy is reducing relapse rates by maintenance therapy—as extensively studied regarding Rituximab in indolent lymphoma or Lenalidomide and other agents in multiple myeloma. However, the results were rather disappointing for DLBCL patients with Rituximab as 1L prolongation (or for R/R DLBCL patients after autologous transplantation) [120,121]. Equally discouraging results were obtained regarding small compounds such as Lenalidomide in the “REMARC” trial or the mTOR inhibitor Everolimus, where no differences in OS were seen [87,120,121,122]. Short-term (i.e., six cycles) maintenance therapy with the PD-L1 blocker Avelumab in the previously described Avelumab-R-CHOP trial will provide preliminary data about the efficacy of ICI for long-term disease control in DLBCL [113]. However, for adequate interpretation of potential clinical benefits due to Avelumab maintenance therapy, additional prospective randomized trials are needed.

Protein kinase C-β (PKC-β) was identified by comparative gene expression profiling as one of 13 prognostically relevant genes related to DLBCL long-term outcome [123]. Inferior prognosis is inferred from high PKC-β expression since it operates as an upstream activator of the PI3K/Akt and NF-κB signaling components of the BCR cascade [124]. Enzastaurin (ENZ), an oral PKC-β and Akt inhibitor, failed to extend OS in the large phase III “PRELUDE” trial as a maintenance therapy for high-risk DLBCL patients (IPI ≥ 3) who achieved a CR after R-CHOP induction [125]. While the sequential administration of ENZ after R-CHOP could not improve OS [126], a new biomarker, termed DGM1 (for De novo Genomic Marker 1, a germline polymorphism on chromosome 8), was found in the subgroup with particularly lasting benefit from ENZ treatment. Accordingly, a new phase III R-CHOP/ENZ trial was initiated based on DGM1 positivity, with first results expected soon [127]. Moreover, ENZ was found to impinge on BTK phosphorylation, leading to enhanced BTK signaling. Accordingly, synergistic antitumor effects of ENZ plus Ibrutinib were explored in preclinical studies [128], with further evaluation to come in clinical trials.

More generally, such a strategy might also motivate investigators from other formally negative phase III studies to revisit their data, retrieve biomarkers of benefitting subgroups and tailor a prospective trial selective enrolling patients based on such a biomarker or gene signature. As alluded to earlier in this section, maintenance treatment in principle has yet to show benefit in DLBCL. An appealing strategy in slowly progressing entities is to suppress outgrowth from a small burden of remaining cells, by extended exposure to an effective component after an effective induction treatment. However, cellular heterogeneity, disease kinetics and post-treatment biologies are different in aggressive lymphomas, leading us to question whether long-term control instead of ultimate elimination of residual disease can serve as an effective strategy. 

## 11. T-Cell-Engaging Therapies Redefine the DLBCL Treatment Landscape

Besides classical immunochemotherapy approaches, bispecific antibodies (BisMAbs) and CAR T-cells are recent, highly promising and rapidly maturing additions to the lymphoma treatment landscape. Several anti-CD19 CAR T-cell products, namely Axicabtagene lisoleucel (Axi-cel), Tisagenlecleucel (Tisa-cel), and Lisocabtagene maraleucel (Liso-cel), were approved over the past few years or may expect registration soon in many parts of the world for the broader indication of R/R DLBCL and aggressive NHL. Several T-cell-engaging BisMAbs with lymphoid surface antigen specificity, to some extent a conceptual CAR T-cell competitor, are currently being exploited in clinical trials. Both BisMAbs and CAR T-cells also suggest themselves as components in 1L therapy.

CAR T-cells reflect apheresed CD4^+^ and CD8^+^ T-cells ex vivo stably transduced with a chimeric antigen receptor (CAR) consisting of an antibody-derived recognition moiety for the target cell-associated surface antigen, here mainly CD19. This chimeric antigen is fused via a costimulatory 4-1BB or CD28 domain to a CD3ζ T-cell activation domain [129]. Axi-cel comes with a CD28 costimulatory domain [130], whereas the Tisa-cel CAR is made of a CD8 hinge region and a 4-IBB costimulatory domain [131,132]. The Liso-cel CAR is similarly designed to the latter and contains, in addition, a nonsignaling/truncated EGFR domain that can be used to control unwanted CAR T-cell expansion in vivo [133]. These engineered autologous anti-CD19 CAR T-cells will be reinfused to the DLBCL patients they were apheresed from under slightly different administration protocols regarding chemoconditioning and dose to exert direct killing of any CD19^+^ cell in the body.

BisMAbs consist of a common binding site shared by the lymphoma population, e.g., CD19 (Blinatumomab) or CD20 (Glofitamab, Mosunetuzumab a.o.) on one hand and the T-cell-specific binding site CD3ε on the other hand. While Blinatumomab, a prototypic early BisMAb, only consists of the two antigen binding sides connected through a linker [134,135], second-generation BisMAbs, e.g., Mosunetuzumab [119,136] or Epcoritamab [137], are structurally full IgG antibodies with two distinct binding specificities (CD3ε and CD20). Glofitamab is an asymmetric IgG antibody consisting of one inner CD3ε binding side and two outer CD20 binding sides, which bind CD20, unlike Mosunetuzumab, at an epitope different from that recognized by Rituximab [138,139,140]. Due to the extremely short half-life of the small (55 kD) single-chain bispecific antibody Blinatumomab, continuous infusions are required over four weeks to achieve effective serum concentrations [141]. For the newer IgG compounds with larger size and a longer half-life, administration every three weeks is sufficient [119,136,138]. The cytotoxic action is thought to be executed upon CD3-mediated T-cell activation via the close proximity of the T-cell to the target cell, ultimately killing it via the perforin/granzyme B pathway [135].

Due to their immune-mediated mechanism of action, all these T-cell-engaging therapies appear especially suitable for R/R DLBCL patients with chemotherapy-refractory disease, as a salvage therapy. Although they are not necessarily restricted to heavily pretreated patients in late lines of therapy, they potentially are a key to improved outcomes for specified high-risk patients, even in 1L. 

### 11.1. Remarkable Long-Term Efficacy of CAR T-Cells in R/R DLBCL Pave Positioning as 1L Component for Specified High-Risk Patients

CAR T-cell products Axi-cel and Tisa-cel are approved by the EMA (and Liso-cel in addition by the FDA) for ≥3L treatment of R/R DLBCL [131,133,142], where they achieve, across the board, a CR in about one-third of the patients three months after administration. While ORRs are profoundly higher, patients only exhibiting a PR as their best response typically gain no lasting disease control. Roughly half of patients experience long-term disease control, most of them based on a CR and continuously detectable, persisting CAR T-cells. Key toxicities are the cytokine release syndrome (CRS), marked by excessive levels of interleukin 6 (IL6) and other pro-inflammatory cytokines in the serum, and neurological events (NE), specifically dubbed ICANS (immune effector cell-associated neurological syndrome). Grade ≥ 3 events occur in 15–25% (CRS) and 10–30% (ICANS) of patients (with somewhat lower frequencies regarding Liso-cel), and typically require management with steroids and/or the IL6 receptor antibody Tocilizumab [143]. Which disease and host factors actually underlie and predict CAR T-cell efficacy is currently under intense investigation. The need for a systemic bridging therapy between T-cell apheresis and CAR T-cell infusion, indicating particularly aggressive disease, was associated with significantly lower OS rates [144,145]. Moreover, the T-cell quality at the time of apheresis is critical for subsequent CAR T-cell function, thereby pointing towards the problem of late T-cell harvesting in heavily pretreated DLBCL patients. CAR T-cell failures may be driven by insufficient propagation of the cells upon reinfusion, loss of the nonessential target antigen CD19 on the lymphoma cells, problematic accessibility of all lymphoma cells especially in bulky disease, and other factors, many yet to be discovered [146].

Three randomized phase III trials recently delivered valuable data for future CAR T-cells as a 2L treatment in a setting of high-risk R/R DLBCL or HGBL patients. Those relapsing within 12 months of induction therapy and eligible for salvage chemotherapy followed by high-dose therapy plus autologous stem cell support as comparator SOC were eligible. Actually, only one-third to one-half of the SOC patients received high-dose therapy, whereas 94% or more of the CAR T-cell-assigned patients were infused. “ZUMA-7” with Axi-cel and “TRANSFORM” with Liso-cel led to significantly improved EFS in favor of CAR T-cells [130,147]. With a short median follow-up period, no significant differences in OS could be observed so far. In contrast, the Tisa-cel-based BELINDA trial in virtually the same setting failed to demonstrate significant differences in EFS [148]. The reasons for this discrepancy are not immediately obvious. One explanation might be that no bridging chemotherapy was permitted in the BELINDA trial, not only potentially selecting for patients with less threatening disease, but more robustly comparing “pure” CAR T-cells vs. an intense sequel of 2L chemotherapies. Moreover, although no head-to-head data between different CAR T-cell products are available, formally not interstudy-comparable PFS data from published ≥ 3L trials suggest that Tisa-cel (median PFS around 2.9 months) may not be as effective as the other two products, Axi-cel (median PFS around 5.9 months) and Lisa-cel (median PFS around 6.8 months) [131,133,142], as also seen in French real-world data about Tisa-cel and Axi-cel [149]. Despite the dismal results of non-CAR T-cell-based 2L concepts so far [150,151,152,153] patients who retain chemosensitivity in 2L appear to have much better outcomes [152].

The ZUMA-12 study is the first to exploit CAR T-cells as a response-adapted escalation in 1L. Specifically, the trial tested Axi-cel-based early consolidation therapy for high-risk DLBCL patients—i.e., *MYC*-involving double-hit cytogenetics or an IPI ≥ 3—who turned out to be iPET-positive after only two cycles of R-CHOP induction; 74% CR and 85% OR rates were observed, with a median PFS due to durable responses not yet being reached, taking into consideration that the median follow-up of 17.4 months is still quite short [154]. While the data are encouraging and pinpoint a potential “solution” for early chemorefractory patients, the decision to change from the planned six cycles SOC towards CAR T-cells based on an insufficient metabolic response as early as two cycles of R-CHOP is ambitious, for some perhaps overambitious. Certainly, a significant proportion of patients with the given entry risk factors and exhibiting metabolic activity in an early iPET would still be cured by mere completion of the SOC. While a subset of patients will clearly benefit from this approach, others will be subjected to overtreatment that comes, in addition, with substantial extra costs. Of special interest are reliable predictors of failure to R-CHOP induction to justify such early intensification. Only a randomized trial can truly determine the potential of CAR T-cells in such a consolidative 1L setting. It must be acknowledged that CAR T-cells have, probably in any line of therapy, great potential for patients with chemotherapy-insensitive disease, as they account for the patient population benefitting least from a 2L high-dose strategy. Hence, markers discriminating those from others being cured by R-CHOP despite a slow initial metabolic response or with a reasonable chance of long-term disease control by salvage 2L chemotherapy followed by high-dose therapy with autologous stem cell transplantation are urgently needed. Perhaps better clinical stratifiers might be PET-positive disease after four cycles (as done in the “GAINED” trial) or EOT, or detectable circulating lymphoma DNA despite a metabolic CR EOT.

Taken together, CAR T-cells are a powerful addition to the therapeutic armamentarium, with immense potential to reshape the therapeutic landscape across all lines of DLBCL treatment. Current data on 1L, 2L and 3L therapies are very promising; however, the effect of preceding rounds of chemotherapy on T-cell quality, the susceptibility of the lymphoma cells to T-cell-mediated killing, and potential evasion mechanisms are problems that need to be solved. Whether CAR T-cells, at least in distinct patient subcohorts, might even entirely replace immunochemotherapy as the initial component of 1L remains to be seen. Moreover, the CAR concept is just at its beginning: multiple CAR specificities propagated in the engineered T-cells, effector cells other than T-cells (i.e., NK cells) and immunologically matched but allogenic cells as a logistically simpler “off-the-shelf” alternative are currently under clinical development.

### 11.2. Bispecific Antibodies—Simpler Logistics, Promising Efficacy and Acceptable Tolerability

Despite its remarkable efficacy in B-cell acute lymphoblastic leukemia (B-ALL), for which it was approved [155], little impact has been seen with Blinatumomab in the R/R DLBCL arena, with a CR rate of only 19% [156]. In contrast, second-generation BisMAbs show very promising results, with lasting remissions and a much more manageable toxicity profile [119,136,138,157,158]. Glofitamab, administered in combination with Obinutuzumab in a heavily pretreated high-risk R/R DLBCL population, achieved CR rates of over 30%, with the mean duration of response (DOR) not reached after a median follow-up time of 13.5 months [138]. First data with Glofitamab or Mosunetuzumab in combination with Polatuzumab in R/R DLBLC and HGBL indicated promising CR rates of around 50% [159,160]. Notably, CRs were even seen with BisMAbs Mosunetuzumab and Odronextamab in patients who relapsed after or were refractory to CAR T-cell therapy [157,159,161]. Additionally, good efficacy in R/R DLBCL could be achieved even though patients received a median of three prior lines of therapy, presumably affecting T-cell quality, underlining the high potential of BisMAbs in this indication [137,138,159]. BisMAb are generally relatively well tolerated, but may, as known from CAR T-cells, account for a CRS. Although CRS is a potentially challenging adverse event, occurring in about half of patients, it became more easily manageable through our growing experience with CAR T-cells, wider clinical use of BisMAbs, step-up dosing of the bispecifics, and specific treatment with steroids and Tocilizumab [162]. Mostly, it is also of low grade. Neurological toxicities, especially ICANs, as frequently faced during early use of Blinatumomab, are rarely seen and mostly mild and self-limiting with the second-generation BisMAbs. 

Mosunetuzumab, a CD20 x CD3 BisMAb, has already been explored in 1L settings, either in combination with CHOP (not R-CHOP due to shared CD20 binding sites) or as a single agent for elderly patients, especially those presenting with compromised performance status or impaired organ function [119,136]. When administered in combination with CHOP for nd DLBCL, CR rates of 85% were achieved, with manageable adverse events observed—all in the range of R-CHOP [136]. Particularly interesting data were reported on Mosunetuzumab as a monotherapy offered to patients over 80 years of age, or over 60 with impaired activity in daily life, or not eligible for standard chemotherapy [119,136]. This higher-risk cohort achieved remarkable and durable responses, with a CR detectable in around half of the patients even before the planned eight cycles were completed, underscoring the high efficacy of this bispecific [119]. To optimize the use of Mosunetuzumab in elderly vulnerable patient populations, prospective trials comparing 1L Mosunetuzumab to current standard therapies such as R-mini-CHOP or R-Bendamustine are needed. So far, the results are promising and more robust data with longer follow-up are expected in the future (see Table 3).

## 12. Discussion

In essence, R-CHOP, administered over six cycles, is an effective 1L therapy for the majority of DLBCL patients irrespective of the underlying molecular heterogeneity. Based on the FLYER phase III study, young patients with no risk factors may be deescalated to only four cycles of the same regimen with no inferior outcome. While relapsing lymphomas are inherently difficult to treat, there are now rapidly expanding options to extend disease control and, in some settings, to enhance survival despite relapse. However, patients refractory to induction therapy or relapsing within the first year face a particularly dismal prognosis [151,152,153,163], further underscoring improved 1L efficacy as the top priority to improve curability. The lack of globally superior efficacy and/or higher toxicity are the ostensible reasons why so many of the previous “R-CHOP ± X” trials failed. Now, the POLARIX trial has demonstrated in a randomized phase III setting a significantly better PFS for the ADC Polatuzumab vedotin + R-CHP arm compared to the R-CHOP standard, thus meeting its primary endpoint [47]. Whether these results will soon and fully replace the R-CHOP standard remains to be seen. Certainly, the POLARIX findings will be, to some extent, practice-changing, and will have profound implications for future trial design in 1L and subsequent lines.

Molecular profiling has not only deepened our biological understanding of this complex disease but has high potential for the future. Trial sponsors and clinical investigators should be motivated to revisit recent randomized phase II and III trials, especially those with an R-CHOP ± X design, to unveil common molecular denominators in benefitting subcohorts (see Table 1). Such a strategy was lately applied to the formally negative PHOENIX trial and unmasked an unparalleled outcome of 100% 3-y EFS for the poorly R-CHOP-responsive MCD subtype under Ibrutinib-R-CHOP [12]. Undoubtedly, such findings require confirmation in prospective studies. Beyond retrospective exploitations of promising agents in completed all-comer or only COO-based preselected trials, prospective “umbrella-style” studies that assign distinct molecular profiles to different treatment options are likely to anticipate the blueprint of future, more personalized decision algorithms. The available results of such strategies at this early stage are modest; the genetic stratifiers rather vague and the candidate agents quite generic [164]. Nevertheless, such a trial design is the prototype for improved personalized treatment decision making in future 1L DLBCL care. A development beyond the R-CHOP vs. Polatuzumab-R-CHP “battle” is remaining vs. new 1L SOC, potentially leading to the end of a single 1L SOC in this disease. Personalized molecularly informed R-CHOP variants or even chemofree induction protocols in 1L will be backed by the 2L standard set by CAR T-cells (and, perhaps, BisMAbs in the future) [130,147]. To some extent, this will buffer the molecular inequalities resulting from a more individualized 1L practice.

Early response assessment is gaining increased attention as an important survey of the clinical and molecular risk factors present at baseline. Specifically, interim FDG-PET imaging, ideally conducted as a tandem metabolic assessment after two or, in particular, four cycles of induction therapy [32]. This reveals the patients at risk of failure because of slow initial response kinetics. Likewise, an insufficient decline in circulating tumor DNA at an interim time point is indicative of poor long-term outcome [165,166,167]. Hence, such personalized response information must be much more routinely obtained to leverage individualized and evidence-based changes in the therapeutic strategy.

Response evaluation of immune oncology-based therapies is different from conventional chemotherapy or targeted agents. So far, we have mostly been thinking of potentially inadequate responses to more classic treatments to consider novel immune oncology strategies as alternatives. Soon, those regimens may enter 1L as sole, pre-phase or combination therapeutics—prompting adapted interpretation of imaging and molecular markers in terms of response assessment to these immune-based treatment options. Likewise, those results can be used as predictors of response before a specific treatment decision has been made. This applies, for instance, to functional judgment of T-cell quality prior to CAR-directed apheresis or BisMAb application. Although our spectrum of therapeutic options operates in the categories of single-agent vs. combination treatments, perhaps in induction plus consolidation or maintenance sequences, too little attention is given to the potential interdependencies agents have in combination. Additionally, different drugs might develop their full potential in smart consecutive sequences, rather than a traditional relapse-triggered sequential use of options.

Criteria outside molecular lymphoma determinants will be of increasing importance in guiding treatment decisions. In addition to the general assessment of biological fitness vs. frailty, specific organ (dys)functions in correspondence to distinct tissue toxicities related to certain agents are of particular importance regarding Anthracycline, Platinum or high-dose eligibility. Similar criteria apply to CAR T-cell or BisMAb suitability. Moreover, our vision of increasingly individually tailored therapies is still highly tumor-centered—given that most of the emerging treatment options either directly engage host immune cells or impinge on stromal bystander and immune cells in the lymphoma microenvironment and system-wide. In the future, such information must be evaluated and converted into clinically relevant, decision-supportive biomarkers [165]. This further applies to the enormous hidden potential of interindividual differences in drug metabolization, whose assessment might allow for the optimization of effective lymphoma drug concentrations in a personalized manner. DLBCL has been recognized as a heterogeneous disease since the beginning of the millennium [7]. Ironically, whether this should trigger molecularly informed, personally tailored concepts or, rather, a molecularly agnostic immune-based attack at surface targets, is a central controversy in the field.

## 13. Concluding Remarks

The transition from the (effective for most) one-size-fits-all approach to a molecularly more refined diagnosis, converted into a lesion-based or biology-driven personalized treatment strategy, will shape the future landscape of DLBCL care. Adjustments based on global risk stratification, in-depth molecular profiling, patient fitness, and early response assessment are critical determinants that guide treatment decisions DLBCL prior to and during 1L therapy (see Table 2 and Table 3). How to adhere to this in today’s practice outside clinical trials, while many important questions remain open and key answers are pending? Our own take on this is to broadly collect clinical and molecular information at baseline, coupled with early PET-based response assessment after three cycles of R-CHOP induction, and judge the biological aggressiveness throughout therapy. As soon as standard treatments appear not to lead to the medical goal anymore, an available, context-specific, agent with anticipated efficacy should be incorporated.

## Figures and Tables

**Table 1 cancers-14-01453-t001:** Clinical trials in DLBCL.

	Trial Name	Treatment	Patients Eligible	Primary Endpoint	Results	Phase	Number of Patients	Clinical Trial Identifier	Reference
First line	FLYER	4× R-CHOP21 vs. 6× RCHOP21	18–60 years, absence of all risk factors (no bulk, normal LDH, Stage I/II disease)	3-y-PFS	4× R-CHOP: less toxicity and noninferior 6× R-CHOP	I/II	592	NCT00278421	[13]
	GOYA	8× G + 6× or 8× CHOP21 vs. 8× R + 6× or 8× CHOP21	>18 years, IPI ≥ 1 or IPI 0 with bulky disease	3-y-PFS	comparable outcome, greater toxicity in G-CHOP (e.g., neutropenia)	III	1414	NCT01287741	[16]
	LNH03-6B study	R-CHOP14 vs. R-CHOP21	60–80 years, IPI ≥1	3-y-EFS	Improved 3-year EFS (56% vs. 60%, respectively)	III	602	NCT00144755	[15]
	DA-EPOCH-R	DA-R-EPOCH vs. R-CHOP	>18 years, stage II–IV disease	EFS	No difference in EFS and OS, higher toxicity in the DA-R-EPOCH	III	524	NCT00118209	[26]
	GAINED	G-CHOP vs. G-ACVBP vs. R-CHOP vs. R-ACVBP	18–60 years, aaIPI 1–3	2-y-EFS	2-y PFS of 83,1% without significant differences in the subgroup, more toxicities in Obinutuzumab arm (e.g., infections Grade 3–5)	III	670	NCT01659099	[32]
	EORTC-20021	8× Gem-R-CHOP21 vs. 8× R-CHOP21	18–70 years, Stage II–IV disease	CR	outcome not improved by addition of Gem, increased toxicity, early closure of the trial	II	25	NCT00079261	[40]
	MAIN	6× RA-CHOP14 + 2×R/8× RA-CHOP21 vs. 6× R-CHOP14 + 2×R/ 8× R-CHOP21	>18 years	PFS/OS until 5 years	Increased cardiotoxicity without increasing efficacy	III	787	NCT00486759	[41]
	HOVON	Lenalidomide-R-CHOP vs. R-CHOP	≥18 y, MYC rearrangement, Ann Arbor stage II-IV	CMR	Safe, results comparable to high intensity chemotherapy regimens	II	85	#2014-002654-39	[42]
	REMoDL-B	R-CHOP-Bortezomib vs. R-CHOP	20–86 years, Stage I–IV disease	PFS	Primary endpoint not reached	III	1076	NCT01324596	[43]
	PHOENIX	Ibrutinib-R-CHOP vs. R-CHOP	>18 years, nonGCB subtype, Stage II-IV disease, IPI ≥ 1, ECOG ≤ 2	EFS	Primary endpoint not reached	III	838	NCT01855750	[44]
	First-MIND	Tafasitamab-R-CHOP vs. Tafasitamab-Lena-lidomide-R-CHOP	≥18 years, IPI 2–5, ECOG 0–2, DH/TH were excluded	safety	Safe, improved ORR (89.7% vs. 93.5%, respectively)	Ib	66	NCT04134936	[45]
	CAVALLI	Venetoclax-R-CHOP vs. R-CHOP	≥18 years, IPI 2–5, ECOG ≤ 2	CR	CR rate improved in BCL2 positive subgroups (specifically BCL2 FISH-positive), higher toxicity in the treatment group (i.e., infection, febrile neutropenia)	Ib–II	208	NCT02055820	[46]
	POLARIX	Pola-R-CHP vs. R-CHOP	18–80 years, IPI 2–5, ECOG ≤ 2	PFS	Improved 2-y PFS, no significant differences in 2-y OS	III	879	NCT03274492	[47]
Currently recruiting	ESCALADE	Acalabrutinib-R-CHOP vs. R-CHOP	18–65 years, Stage II–IV disease, R-IPI 2–5	PFS	Currently recruiting	III	Planned 600	NCT04529772.	[48]
	ImbruVeRCHOP	Ibrutinib + Bortezomib + R-CHOP	≥60 years, IPI ≥ 2	2-y-PFS	Currently recruiting	I/II	Planned 34	NCT03129828	[49]

Selection of 1L studies in DLBCL with treatment additions to the SOC R-CHOP: upper section, completed trials; bottom section, two studies still actively recruiting. Abbreviations: y: year, PFS: progression-free survival, CR: complete response, CMR: complete metabolic response, EFS: event-free survival, R: Rituximab, CHOP: combination of Cyclophosphamide, Doxorubicin, Vincristine and Prednisolone, R-CHP: R-CHOP without Vincristine, R-CHOP14: R-CHOP every 14 days, R-CHOP21: R-CHOP every 21 days, DA-R-EPOCH: dose-adjusted R-CHOP plus Etoposid, Pola: Polatuzumab vedotin, Gem: Gemcitabine, G: Obinutuzumab, A: Bevacizumab, IPI: international prognostic index, aaIPI: age-adjusted IPI, DH: double hit, TH: triple hit.

**Table 2 cancers-14-01453-t002:** Chemotherapy-free induction therapy regimens in DLBCL.

Trial	Treatment	Patients Eligible	Primary Endpoint	Phase	Included Number of Patients	Results	Clinical Trial Identifier	Reference
GO40554	Mosunetuzumab	≥80 y or ≥60 y and not eligible for standard therapy	Safety	I/II	40	Safe, ORR 68%, CR 42%	NCT03677154	[119]
SMART-Start	2 cycles of RIL followed by R-CHOP	≥18 y, ECOG 0–2, non-GCB,	ORR after 2 cycles RIL, CRR after RIL and R-CHOP	II	60	After 2xRIL ORR 86%, CRR 36%	NCT02636322	[117]
Avr-CHOP	2 cycles R-Avelumab followed by R-CHOP	≥18 y, ECOG 0–2, stage II–IV disease	Safety	II	28	After 2 cycles of R-Avelumab ORR 60%, CMR 21%	NCT03244176	[113]

Selection of chemotherapy-free 1L regimens in DLBCL. Abbreviations: y: year, CR: complete response, CRR: complete response rate, CMR: complete metabolic response, ORR: overall response rate, R: Rituximab, CHOP: combination of Cyclophosphamide, Doxorubicin, Vincristine and Prednisolone, ECOG: eastern cooperative oncology group performance status, GCB: germinal center B-cell, RIL: combination of Rituximab, Ibrutinib and Lenalidomide.

**Table 3 cancers-14-01453-t003:** Ongoing clinical trials investigating bispecific antibodies.

	Bispecific Antibody, Binding Sides	Treatment	Patients Eligible	Primary Endpoint	Phase	Planned Number of Patients	Estimated Completion Date	Clinical Trial Identifier
Currently recruiting	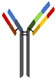	CD3 x dualCD20Glofitamab	Glofitamab-R-CHOP or Glofitamab-Pola-R-CHP	18–65 y, IPI ≥ 3 ECOG 0–1, untreated high-risk DLBCL	Safety, dose finding	Ib/II	80	07/25	NCT04914741
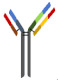	CD3 x dualCD20Glofitamab	Glofitamab-R- or G-CHOP	≥18 y, ECOG 0–3, untreated or R/R NHL	Dose-limiting toxicities	Ib	172	12/23	NCT03467373
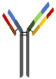	CD3 x dualCD20Glofitamab	Glofitamab-R-CHOP	≥18 y, IPI1–5, ECOG 0–2, Circulating tumor DNA high-risk DLBCL	CR	II	40	12/24	NCT04980222
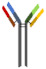	CD3 x CD20Odronextamab	Odronextamab-Mono	≥18 y, ECOG ≤1, CD20 pos B-cell malignancies	Safety, ORR	I	256	12/25	NCT02290951
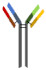	CD3 x CD20Epcoritamab	Epcoritamab + R-CHOP or R-Lena or R-Benda or GemOx or R-DHAX/C	≥18 y, ECOG 0–2, B-NHL	Safety, preliminary antitumor activity	Ib/II	130	09/24	NCT04663347
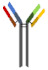	CD3 x CD20Mosunetuzumab	Mosunetuzumab or Mosunetuzumab-Polatuzumab vedotin	≥18 y, ECOG 0–2, DLBCL with SD, PR or CR after induction therapy	Safety, CR, ORR	I/II	188	09/25	NCT03677154
closed	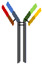 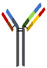	Mosunetuzumab & Glofitamab	Glofitamab- or Mosunetuzumab-GemOx	≥18 y, ECOG 0–2, R/R NHL	Safety	Ib	23	Completed	NCT04313608

Ongoing clinical trials investigating bispecific antibodies including schematic illustration of the antibody structure and its binding sides. Colors used: Dark and light green/blue: heavy chain; red, brown, yellow and dark yellow: light chain; gray: Fc region. Abbreviations: y: year, SD: stable disease, PR: partial remission, CR: complete response, ORR: overall response rate, R: Rituximab, CHOP: combination of Cyclophosphamide, Doxorubicin, Vincristine and Prednisolone, Gem: Gemcitabine, Ox: Oxaliplatin, G: Obinutuzumab, R/R: relapsed/refractory, NHL: non-Hodgkin’s lymphoma, DLBCL: diffuse large B-cell lymphoma, ECOG: eastern cooperative oncology group performance status, IPI: international prognostic index.

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
