# Peer review of "DLBCL 1L—What to Expect beyond R-CHOP?"

_cancers, 2022, doi:10.3390/cancers14061453_

Round 1

Reviewer 1 Report

The updated manuscript reads much better and all points have been sufficently addressed by the authors.

Especially the elaborations on the POLARIX trial are stimulating.

Table 1: There is a bracket missing in the FLYER row, patients eligible collumn

Line 287: While L265P is the AA change which is overwhelmingly given in the literature, the correct MANE-transcript reports the change as L252P and in COSMIC the AA change is given as L273P. So finding the alteration in a database is tricky by only giving the L265P. I would maybe remain descriptive by writing: Activating mutations in the TIR-domain of MyD88 ...

Author Response

Reviewer 1: Minor

The updated manuscript reads much better and all points have been sufficiently addressed by the authors. Especially the elaborations on the POLARIX trial are stimulating.

We are pleased by the referee’s very positive judgment of our fundamentally revised manuscript.

Table 1: There is a bracket missing in the FLYER row, patients eligible column.

Thanks for pointing out. Now corrected.

Line 287: While L265P is the AA change which is overwhelmingly given in the literature, the correct MANE-transcript reports the change as L252P and in COSMIC the AA change is given as L273P. So finding the alteration in a database is tricky by only giving the L265P. I would maybe remain descriptive by writing: Activating mutations in the TIR-domain of MyD88

We agree that there seems to be an issue with proper AA counting in MyD88 in the literature. The referee’s suggestion is a smart way to circumvent the problem – we changed the text accordingly.

Reviewer 2 Report

The authors extensively reviewed the paper.

All points raised have been appropriately addressed.

Author Response

We thank this referee for another round of reviewing. No concerns remain. 

This manuscript is a resubmission of an earlier submission. The following is a list of the peer review reports and author responses from that submission.

Round 1

Reviewer 1 Report

Stegemann and colleagues provide an up-to-date overview of novel first-line treatment strategies in DLBCL. The description of individual trials is detailed and concise. The tabular summary of clinical studies is very informative and well structured. My points to further strengthen the manuscript are:

  • Briefly describe what DLBCL is in a short introductory paragraph with a few words on incidence, age distribution, established biomarkers/problems in routine diagnostics (from paragraphs 67-76 & 77-85) and heterogeneity of DLBCL/HG-BNHL, especially with regard to clinical behavior. This may serve to illustrate the rationale for different strategies in the presented trials.

  • Line 72: Lymph2Cx is one of multiple GEP assays. For instance, the authors discuss the PHOENIX trial which used a different GEP platform (HTG). Please mention that there are 1) multiple IHC algorithms and 2) multiple GEP techniques, none of which has been agreed on as the best method to characterize COO.

  • Line 79: Double expressor DLBCL refers to an increased expression of Myc and BCL2. Add “clinically assessed by IHC”. Add PMID: 22665537.

  • Table 1: Population seems to refer to the study population. While I agree that this is more informative than inclusion criteria, it should be stated more clearly – also see next point.

  • Table 1: Inclusion criteria for the FLYER trial was based on age adjusted IPI (specifically, absence of all risk factors) and absence of bulky disease. The few intent-to-treat participants with IPI 1 were retrospectively reclassified. I suggest adding that only 1% of the intent-to-treat population had an IPI of 1, otherwise the reader might conclude that 4 cycles of R-CHOP are non-inferior for all IPI 1 patients. Also adapt Line 40.

  • Meanwhile the POLARIX trial has been published. I suggest updating the paragraph and Table1. Please update other studies, if necessary, especially in light of ASH annual meeting 2021.

  • Add a reference for the statement in Line 296: “patients refractory to induction therapy or relapsing within the first year face a dismal prognosis” – e.g., SCHOLAR-1 trial (PMID: 28774879)

  • The paper starts out with Introduction but has no subheadings. Those could be helpful in structuring the trials (e.g.: R-CHOP + X, Chemofree Inductions, Biologicals, Cellular Therapies). At least a heading for the discussion (Line 295) should be added. Dismiss this point, if journal requirements do not allow for headings.

  • Sentences tend to be long and sometimes convoluted. Proof-reading by a native English speaker will help to reduce wordy passages and to smoothen the language. Try to limit the amount of information within a given sentence. Try to use shorter sentences and divide sentences. Some suggestions:

Line 46: “The extension of R-CHOP […] was actually established in the pre-Rituximab era.” might read better as “Etoposide is being added to (R-)CHOP in high-risk patients since before the Rituximab era.”

Line 159: “Nevertheless, […] GCB-subtype patients were enrolled.” – “Reevaluation of COO-subtype by GEP unveiled a GCB subtype in 15-19% of the study population. Hence patients below 60 experienced improved outcome irrespective of subtype.”

Line 224-227: “Besides classical immunochemotherapy approaches, bispecific antibodies (BisMAbs) and chimeric-antigen receptor (CAR) T-cells are recent and highly promising additions to the lymphoma treatment landscape, especially in chemotherapy-refractory disease, and, hence, not necessarily restricted to heavily pretreated patients in late lines of therapy.” – “Bispecific antibodies (BisMAbs) and chimeric-antigen receptor (CAR) T-cells are highly promising additions to the lymphoma treatment landscape, especially in early chemotherapy-refractory disease”

Author Response

Point-by-point reply

Stegemann and colleagues provide an up-to-date overview of novel first-line treatment strategies in DLBCL. The description of individual trials is detailed and concise. The tabular summary of clinical studies is very informative and well structured.

  • We are pleased by the referee’s very positive judgment of our contribution and hereby express our gratitude for the time invested to comment on our work. We found the statements of great help to further improve the manuscript.

My points to further strengthen the manuscript are:

Briefly describe what DLBCL is in a short introductory paragraph with a few words on incidence, age distribution, established biomarkers/problems in routine diagnostics (from paragraphs 67-76 & 77-85) and heterogeneity of DLBCL/HG-BNHL, especially with regard to clinical behavior. This may serve to illustrate the rationale for different strategies in the presented trials.

  • We thank the referee for pointing this out. Although intended as part of a series of papers on DLBCL, an introductory paragraph emphasizing the general epidemiological and clinical background of this entity, and touching on the heterogeneity as a critical determinant of unsatisfying outcome in greater depth is a very good suggestion to pave the ground for the subsequent discussion of future treatment options beyond the R-CHOP standard. We revised the manuscript accordingly.

Line 72: Lymph2Cx is one of multiple GEP assays. For instance, the authors discuss the PHOENIX trial which used a different GEP platform (HTG). Please mention that there are 1) multiple IHC algorithms and 2) multiple GEP techniques, none of which has been agreed on as the best method to characterize COO.

  • We truly appreciate this important point – as it indicates technical difficulties to address COO, divergent ways to come to a COO designation, and the lack of a COO reference, since the COO itself is a transcriptome-inaugurated concept, not a biologically defined status (Alizadeh-AA et al., Nature, 2000). IHC-based algorithms (Hans et al, 2004, Blood; Choi et al, 2009, AACR, Visco-C et al, Leukemia, 2012) are generally understood as less accurate surrogate markers for more complex transcriptome-based assays and their underlying technical platforms (such as NGS, NanoString, Illumina DASL [cDNA-mediated Annealing, Selection, extension and Ligation], HTG and others) (De Jong-D et al., JCO, 2007; Meyer-PN et al., JCO, 2011; Visco-C et al., Leukemia, 2012; Scott-DW et al., Blood, 2014; Scott-DW et al., JCO, 2015) . We elaborated on this aspect in the novel version of the manuscript.

Line 79: Double expressor DLBCL refers to an increased expression of Myc and BCL2. Add “clinically assessed by IHC”. Add PMID: 22665537.

  • The immunohistochemical “Double Hit-Score” introduced by Tina Marie Green et al. has been clinically validated in a retrospective manner (JCO, 2012 – reference now added). Despite its potentially misleading denomination – not referring to “double hit” lymphomas that carry Myc and Bcl2 or Bcl6 gene translocations – it is certainly not undisputed. There are by now a plethora of papers addressing the prognostic impact of Myc and Bcl2 (over)expression in DLBCL with and without underlying translocations, with the double-expressors as compared to the double-hits being less clearly clinically inferior under standard R-CHOP immunochemotherapy. We elaborated on this discussion and rephrased the corresponding section in the revised version accordingly.

Table 1: Population seems to refer to the study population. While I agree that this is more informative than inclusion criteria, it should be stated more clearly – also see next point.

  • Agreed, this might be misleading. The term “Population” indeed refers to the patient population eligible, e. to some of the inclusion criteria. We renamed to “Patients eligible”.

Table 1: Inclusion criteria for the FLYER trial was based on age adjusted IPI (specifically, absence of all risk factors) and absence of bulky disease. The few intent-to-treat participants with IPI 1 were retrospectively reclassified. I suggest adding that only 1% of the intent-to-treat population had an IPI of 1, otherwise the reader might conclude that 4 cycles of R-CHOP are non-inferior for all IPI 1 patients. Also adapt Line 40.

  • We apologize for the mistake. Entry criteria for the FLYER study were an aaIPI of 0 (not 1), and, indeed, no bulk. We changed the information for FLYER in the new “Patients eligible” category to “18-60 y, absence of all risk factors (no bulk, normal LDH, stage I/II) and further specified this point in the text of the revised manuscript.

Meanwhile the POLARIX trial has been published. I suggest updating the paragraph and Table1. Please update other studies, if necessary, especially in light of ASH annual meeting 2021.

  • Yes, indeed, the POLARIX phase III trial, positive in terms of its primary endpoint PFS, has been published meanwhile (Tilly-H et al., NEJM, 2021). We updated table and text regarding this important information, and included other trials in light of the recent ASH meeting, where applicable.

Add a reference for the statement in Line 296: “patients refractory to induction therapy or relapsing within the first year face a dismal prognosis” – e.g., SCHOLAR-1 trial (PMID: 28774879)

  • We thank the referee for the suggestion – and cited now (in addition to the CORAL study [Gisselbrecht-C et al., JCO, 2010]) the SCHOLAR-1 trial (Crump-M et al., Blood, 2017) as suggested plus the Hamadani-M et al. investigation (Biol Blood Marrow Transplant, 2014), which adds to the dismal prognosis of early R-CHOP-failing patients if they present as chemoinsensitive to subsequent salvage therapy.

The paper starts out with Introduction but has no subheadings. Those could be helpful in structuring the trials (e.g.: R-CHOP + X, chemotherapy free Inductions, Biologicals, Cellular Therapies). At least a heading for the discussion (Line 295) should be added. Dismiss this point, if journal requirements do not allow for headings.

  • We thank the reviewer to point this out. The structure of the entire text is now revised and improved by subheadings as suggested by the referee. In this context, we also exchanged tables 2 and 3 for improved readability.

Sentences tend to be long and sometimes convoluted. Proof-reading by a native English speaker will help to reduce wordy passages and to smoothen the language. Try to limit the amount of information within a given sentence. Try to use shorter sentences and divide sentences. Some suggestions:

Line 46: “The extension of R-CHOP […] was actually established in the pre-Rituximab era.” might read better as “Etoposide is being added to (R-)CHOP in high-risk patients since before the Rituximab era.”

Line 159: “Nevertheless, […] GCB-subtype patients were enrolled.” – “Reevaluation of COO-subtype by GEP unveiled a GCB subtype in 15-19% of the study population. Hence patients below 60 experienced improved outcome irrespective of subtype.”

Line 224-227: “Besides classical immunochemotherapy approaches, bispecific antibodies (BisMAbs) and chimeric-antigen receptor (CAR) T-cells are recent and highly promising additions to the lymphoma treatment landscape, especially in chemotherapy-refractory disease, and, hence, not necessarily restricted to heavily pretreated patients in late lines of therapy.” – “Bispecific antibodies (BisMAbs) and chimeric-antigen receptor (CAR) T-cells are highly promising additions to the lymphoma treatment landscape, especially in early chemotherapy-refractory disease”

  • We are particularly grateful for the extra effort this referee undertook to enhance clarity and crisp readability. We not only followed the specific suggestions made in the examples above but revised the entire text in this regard.
  • We fundamentally revised the entire manuscript to focus on every single point of your very helpful comments.

Reviewer 2 Report

This is a relatively interesting review dealing with the first-line therapy of DLBCL.   The authors have analyzed in detail a series of studies providing different novel options for the treatment of this heterogeneous disease.

However, the paper needs important changes.

I find that the present version does not have any structure.

The authors should be asked to better organize and structure the paper sharing the manuscript in different chapters. This will make the manuscript more readable.

According to the title, authors promise to give their personal opinion on what we expect beyond R-CHOP. 

However, in the present version of the paper, they avoid providing their personal point of view on the potential future development of DLBCL therapy. 

Authors should further expand and clarify the part dedicated to the heterogeneity of DLBCLs.

For instance, high-grade B-cell lymphoma (HGBL) cases are now considered a distinct entity in the 2016 World Health Organization Lymphoid Tumor Classification.  This is a subset of aggressive cases that, despite overlapping clinical and pathological characteristics, are different from  DLBCL.  The category includes lymphomas that were previously named “Burkitt-like” and “high grade” as well as “double-hit” or “triple-hit.”

In addition to the major category “HGBL with MYC and BCL2 and/or BCL6 translocations,” there is a subset called “HGBL-not otherwise specified” (NOS).

 This updated categorization should be considered in the clinic, as HGBLs are more aggressive than DLBCL and require distinct therapeutic approaches. 

As rightly reported by the authors, CARTs have changed the treatment landscape for R/R aggressive B-cell lymphomas. Results of ZUMA-12 indicate that therapy with CARTs can be anticipated thus providing a sort of "early consolidation" to patients with an interim PET  positive after two cycles of R-CHOP.  This is an interesting and innovative approach that needs to be further commented.

Authors try to make a comparison between CARTs and bispecific monoclonal antibodies.

Some comments relative to the cost-effectiveness of these procedures should be done taking into account that CART is a single-shot procedure.

Author Response

Point-by-point reply

This is a relatively interesting review dealing with the first-line therapy of DLBCL.   The authors have analyzed in detail a series of studies providing different novel options for the treatment of this heterogeneous disease.

However, the paper needs important changes.

I find that the present version does not have any structure.

The authors should be asked to better organize and structure the paper sharing the manuscript in different chapters. This will make the manuscript more readable.

  • We like to thank this referee for his/her positive overall impression and valuable comments, which are truly helpful and highly appreciated. We added multiple subheadings to better structure the text. In this context, we also exchanged tables 2 and 3 for improved readability. We have taken another critical look at the manuscript and elaborated a bit further, as well as updating it with additional research findings.

According to the title, authors promise to give their personal opinion on what we expect beyond R-CHOP. 

However, in the present version of the paper, they avoid providing their personal point of view on the potential future development of DLBCL therapy. 

  • Unlike assumed, this review is not really an “opinion” manuscript. It primarily summarizes recent study results and emerging developments to improve the 1L treatment landscape in this heterogeneous entity. Hence “…what to expect beyond R-CHOP” is less of an authors’ opinion-based future-telling but reflects objective activities visible today to impact clinical practice in this space tomorrow. Nevertheless, the Discussion and Concluding Remarks sections is, of course, driven by the authors’ personal anticipation. We have sharpened this aspect further in the revised version of the manuscript.

Authors should further expand and clarify the part dedicated to the heterogeneity of DLBCLs.

For instance, high-grade B-cell lymphoma (HGBL) cases are now considered a distinct entity in the 2016 World Health Organization Lymphoid Tumor Classification.  This is a subset of aggressive cases that, despite overlapping clinical and pathological characteristics, are different from DLBCL.  The category includes lymphomas that were previously named “Burkitt-like” and “high grade” as well as “double-hit” or “triple-hit.”

  • We made clear, throughout the manuscript, that heterogeneity – in addition to the two thirds of DLBCL cases being cured by R-CHOP induction – accounts for the tremendous challenge to improve outcome to 1L therapy. While HGBL with Myc and Bcl2 and/or Bcl6 translocations are commonly viewed as a particularly dismal subset, this is less clear with HGBL-NOS and certainly highly disputed regarding Myc/Bcl2 double-expressor lymphomas lacking “double-hit” translocations. Likewise, it is only fair to say that no intensified treatment strategy, so far, demonstrated superior outcome for HGBL to standard R-CHOP in a randomized trial. The emerging data from the ZUMA-12 single-arm trial (Neelapu-SS,… …Chavez-JC, ASH Abs. 405 2020) to demonstrate a role for Axi-Cel CAR T-cells as consolidative extension of insufficient induction as judged by positive interim PET after two cycles of therapy provide a signal, but not necessarily the solution for these hard-to-cure patients, since many of the CAR T-cell recipients would still have achieved a lasting remission if completing the standard of care. We agree that this point is interesting and currently under intense debate in the field – and, hence, have expanded on this discussion in the revised version of the manuscript.

In addition to the major category “HGBL with MYC and BCL2 and/or BCL6 translocations,” there is a subset called “HGBL-not otherwise specified” (NOS).

  • See above. We also like to clarify that we did not write up a comprehensive review of all HGBL/DLBCL subentities as listed in the most recent WHO classification with respect to subtype-specific induction therapies. Rather, it should be clear that no pathology subtype-specific 1L treatment stratification has been established yet, nor is it clearly in sight for other risk determinants (beyond a distinct WHO subtype) at diagnosis. Again, the revised version gives this point somewhat more room.

 This updated categorization should be considered in the clinic, as HGBLs are more aggressive than DLBCL and require distinct therapeutic approaches. 

  • See above; there is no change for HGBL regarding the standard-of-care yet; while the enhanced risk is consented among the experts in the community, superior treatment alternatives are not. As stated above, we have now expanded on this important point and its emerging perspectives for future treatment strategies.

As rightly reported by the authors, CARTs have changed the treatment landscape for R/R aggressive B-cell lymphomas. Results of ZUMA-12 indicate that therapy with CARTs can be anticipated thus providing a sort of "early consolidation" to patients with an interim PET positive after two cycles of R-CHOP.  This is an interesting and innovative approach that needs to be further commented.

  • We thank the referee for his/her request to further comment on early consolidation treatment with CAR T-cells for high-risk patients (double-hit and beyond) with a positive interim PET. As stated above, we present at balanced view on this approach, as it bears the possibility for overtreatment and might give a wrong impression of a specifically good outcome after CAR T-cell therapy. This discussion is of particular interest in light of the three ASH 2021-presented 2L CAR T-cell trials, of which two (ZUMA-7 and TRANSFORM) were positive and one (BELINDA) was negative. We also like to remind us of the data by Hamadani-M et al. (Biol Blood Marrow Transplant, 2014), which show a much better outcome to salvage therapy (then assumed by the CORAL study [Gisselbrecht-C et al., JCO, 2010] and the SCHOLAR-1 trial [Crump-M et al., Blood, 2017], not necessarily inferior to CAR T-cells), if R/R patients exhibit chemosensitive disease in response to salvage chemotherapy. We have touched on this discussion to a certain extent in the revised version of the manuscript, but like to emphasize that these complex 2L reflections are not yet ready to feedback on 1L decisions.

Authors try to make a comparison between CARTs and bispecific monoclonal antibodies. Some comments relative to the cost-effectiveness of these procedures should be done taking into account that CART is a single-shot procedure

  • We thank the referee for his/her input on comparability in terms of cost-effectiveness when comparing single and multiple dose applications. We first and foremost focused on the medical benefits and the underlying mechanism of action, efficacy, and tolerability. The economic aspect, as important as it is from the public health perspective, cannot be a prime topic, as long as treatment-line positioning and long-term benefits of these different but similar cell-based immune-oncologicals have been determined. As long as 50% of the CAR T-cell recipients don’t achieve lasting benefit, additional treatment lines including bispecific antibodies are needed; CAR T-cells may also be repeatedly administered, as currently being investigated, or one CAR T-cell approach might be followed by another CAR T-cell approach of different specificity – collectively all arguments not to discuss costs in a 1L review at this point.
  • We fundamentally revised the entire manuscript to focus on every single point of your very helpful comments.
